# Zika seroprevalence declines and neutralizing antibodies wane in adults following outbreaks in French Polynesia and Fiji

Alasdair D Henderson[1†], Maite Aubry[2†], Mike Kama[3,4], Jessica Vanhomwegen[5], Anita Teissier[2], Teheipuaura Mariteragi-Helle[2], Tuterarii Paoaafaite[2], Yoann Teissier[6], Jean-Claude Manuguerra[5], John Edmunds[1], Jimmy Whitworth[1], Conall H Watson[1], Colleen L Lau[7], Van-Mai Cao-Lormeau[2‡*], Adam J Kucharski[1‡*]

[1]Centre for the Mathematical Modelling of Infectious Diseases, London School of Hygiene and Tropical Medicine, London, United Kingdom; [2]Institut Louis Malardé, Papeete, French Polynesia; [3]Fiji Centre for Communicable Disease Control, Suva, Fiji; [4]The University of the South Pacific, Suva, Fiji; [5]Institut Pasteur, Paris, France; [6]Direction de la Santé de la Polynésie française, Papeete, French Polynesia; [7]Australian National University, Canberra, Australia

*For correspondence:
mlormeau@ilm.pf (V-MC-L);
adam.kucharski@lshtm.ac.uk (AJK)

[†]These authors contributed equally to this work
[‡]These authors also contributed equally to this work

Competing interests: The authors declare that no competing interests exist.

**Abstract** It has been commonly assumed that Zika virus (ZIKV) infection confers long-term protection against reinfection, preventing ZIKV from re-emerging in previously affected areas for several years. However, the long-term immune response to ZIKV following an outbreak remains poorly documented. We compared results from eight serological surveys before and after known ZIKV outbreaks in French Polynesia and Fiji, including cross-sectional and longitudinal studies. We found evidence of a decline in seroprevalence in both countries over a two-year period following first reported ZIKV transmission. This decline was concentrated in adults, while high seroprevalence persisted in children. In the Fiji cohort, there was also a significant decline in neutralizing antibody titres against ZIKV, but not against dengue viruses that circulated during the same period.

## Introduction

Zika virus (ZIKV), a *Flavivirus* primarily transmitted to humans by *Aedes* mosquitoes, was first reported in the Pacific region on Yap island (Federated States of Micronesia) in 2007 (*Duffy et al., 2009*). Six years later, there was a large ZIKV outbreak in French Polynesia (*Cao-Lormeau et al., 2014*) where an estimated 11.5% of the population visited healthcare facilities with clinical symptoms suggestive of ZIKV infection (*Kucharski et al., 2016*). Since then the virus has spread across the Pacific region (*Musso et al., 2014*), including to Fiji where cases of ZIKV infection were first detected in July 2015 (*World Health Organisation, 2015*). The same year, cases of ZIKV infection in Latin America were reported for the first time (*Zammarchi et al., 2015*). From February 1 to November 18, 2016, due to its rapid spread and association with birth defects, microcephaly in newborns and Guillain-Barré syndrome in adults (*Cao-Lormeau et al., 2016*) the WHO declared ZIKV a Public Health Emergency of International Concern (*World Health Organisation, 2016*). At the end of 2016, outbreaks had declined in most of the countries recently affected (*O'Reilly et al., 2018*). However, ZIKV was still circulating in 2018 in several countries, including Fiji and Tonga in the Pacific region (*World Health Organisation, 2019*).

In countries with known ZIKV outbreaks, the few serological surveys that have been published found a high level of ZIKV seroprevalence following the outbreak. In French Polynesia, a population-

**eLife digest** Since the Zika virus first emerged in the Pacific Islands in 2007, it has caused many outbreaks in the Pacific and Latin America. Some scientists thought that after exposure to the virus people would develop long-term immunity to it, reducing the number of outbreaks in the future. Several studies supported this idea. These studies showed that many people recently infected with Zika developed antibodies in their blood that might protect them from becoming ill during future outbreaks. But it was not clear how long this protection would last.

To better understand how immunity to the Zika virus changes over time, Henderson, Aubry et al. combined data from eight surveys that collected blood samples at different time points during Zika outbreaks in French Polynesia and Fiji. The analysis showed that the proportion of people with detectable antibodies against the Zika virus increased in both countries after the outbreaks. In children these immune responses persisted for years, but antibody levels declined over time in adults. By contrast, antibodies to the closely related dengue virus did not wane over time in individuals tested for both viruses in Fiji in 2013, 2015 and 2017.

The data suggest that immunity against the Zika virus may not last as long as previously thought, which could affect the chances of future outbreaks. The findings may also have implications for researchers studying the virus, because the number of people with antibodies against the virus is not a good estimate of how many people were initially infected. More studies are needed to understand immunity to Zika virus over time and how it may affect future outbreaks.

representative cross-sectional serological survey at the end of the outbreak in 2014 found a seroprevalence of 49% (*Aubry et al., 2017*). In Martinique, a study of blood donors showed a post-outbreak seroprevalence of 42% in 2015 (*Gallian et al., 2017*). In Salvador, Northeastern Brazil, a serosurvey in 2016 of prospectively sampled individuals including microcephaly and non-microcephaly pregnancies, HIV-infected patients, tuberculosis patients, and university staff, found a post-outbreak seroprevalence of 63% (*Netto et al., 2017*). Another study in Salvador, conducted in a longterm health cohort, also found a post-outbreak seroprevalence of 63% (*Rodriguez-Barraquer et al., 2019*). Finally, in paediatric and household cohort studies in Managua, Nicaragua, ZIKV seroprevalence was estimated to be 46% in households following the outbreak in 2016 (*Zambrana et al., 2018*).

It has been suggested that infection with ZIKV confers immunity that lasts several years; if so, the high level of seroprevalence in affected countries may reflect sufficient herd immunity for the current ZIKV epidemic to be over in many locations, with the virus unable to re-emerge for decades to come (*Kucharski et al., 2016*; *O'Reilly et al., 2018*; *Netto et al., 2017*; *Ferguson et al., 2016*). Recent evidence suggests that neutralizing antibodies can distinguish between ZIKV and dengue virus (DENV) – a closely related *Flavivirus* – and that the immune response following ZIKV infection can persist over a year (*Montoya et al., 2018*; *Griffin et al., 2019*). It has also been suggested that primary ZIKV infection may confer protective immunity (*Osuna et al., 2016*). However, ZIKV serosurveys conducted at the end of the outbreak in French Polynesia and 18 months later found a drop in seroprevalence in the Society Islands, the archipelago where over 85% of the inhabitants of French Polynesia reside (*Aubry et al., 2017*). Therefore, the long-term antibody response following a ZIKV outbreak remains unclear.

Here, we explore short- and long-term seroprevalence against ZIKV as well as neutralizing responses against ZIKV following two ZIKV outbreaks in the Pacific region. We compared results from five serological surveys in the Society Islands, French Polynesia, over a seven-year period, and three serial serological surveys in the same cohort of individuals in Central Division, Fiji, over a fouryear period. These surveys span the pre- and post- outbreak period in each country, allowing us to examine temporal changes in antibody responses following a ZIKV outbreak.

## Results

In French Polynesia, seroprevalence of IgG antibodies against domain III of the ZIKV envelope glycoprotein in blood donors recruited before October 2013 was <1% (0.3–2%), which confirmed that the virus had not previously circulated in the population (*Table 1*). Analysis of samples collected in the

**Table 1.** Seroprevalence of ZIKV among participants in five serological surveys in French Polynesia and three serological surveys in Fiji, conducted between July 2011 and June 2018.

| Date | Country | Population and assay used | Age range (median) | Total no. seropositive/total no. tested | Seroprevalence % [95% CI] |
|---|---|---|---|---|---|
| French Polynesia - General Population | | | | | |
| Jul 2011-Oct 2013 | Society Islands, French Polynesia | Blood donors, ELISA | 18–75 (36) | 5/593 | 0.8 [0.3–2.0] |
| Nov 2013 | *First confirmed local transmission of ZIKV in French Polynesia* | | | | |
| Feb-Mar 2014 | Society Islands, French Polynesia | General, ELISA | 13–77 (47) | 18/49 | 37 [26-47]* |
| Sep-Nov 2015 | Society Islands, French Polynesia | General, MIA | 4–88 (43) | 154/700 | 22 [16-28]* |
| French Polynesia - schoolchildren | | | | | |
| May-Jun 2014 | Society Islands, French Polynesia | School children, ELISA | 6–16 (11) | 312/476 | 66 [60-71]* |
| Jun-2018 | Society Islands, French Polynesia | School children, MIA | 6–16 (11) | 291/457 | 64 [58-69]* |
| Fiji | | | | | |
| Oct-Nov 2013 | Central Division, Fiji | General, MIA | 2–78 (24) | 12/189 | 6.3 [3.3–11] |
| Jul 2015 | *First confirmed local transmission of ZIKV in Fiji* | | | | |
| Nov-2015 | Central Division, Fiji | General, MIA | 4–80 (26) | 45/189 | 24 [18-31] |
| Jun-2017 | Central Division, Fiji | General, MIA | 6–82 (28) | 23/189 | 12 [7.9–18] |

* CIs were calculated taking into account the cluster sampling design (**Aubry et al., 2017**) and using the Fisher exact test.

MIA – microsphere immunoassay.

general population of the Society Islands of French Polynesia after the emergence of ZIKV showed a decrease in ZIKV seroprevalence from 37% (26–47%) to 22% (16–28%) between February-March 2014 and September-November 2015 (chi-squared test, p=0.03). In Fiji, analysis of the serum samples serially collected from a cohort of participants in the Central Division showed an increase in ZIKV seroprevalence from 6.3% (3.3–11%) in October-November 2013 to 24% (18–31%) in November 2015 (chi-squared test, p<0.0001), and then a decrease to 12% (7.9–18%) by June 2017 (chi-squared test, p=0.005). In this cohort, based on IgG results tested by microsphere immunoassay (MIA), 6 of the 189 participants seroconverted (from negative to positive) and 28 seroreverted (from positive to negative) to ZIKV between 2015 and 2017 (McNemar's test, p=0.0003).

To investigate possible factors influencing the decline in seroprevalence, we compared the seroprevalence profiles in children (defined as ≤16 years) and adults (>16 years) in both settings (*Table 1* and *Figure 1*). In French Polynesia, although ZIKV seroprevalence declined in the general population from the Society Islands over 18 months, there was no evidence of a significant decline in seroprevalence in two serosurveys conducted four years apart in schoolchildren aged 6 to 16 years, with 66% (60–71%) positive in 2014 and 64% (58–69%) in 2018 (chi-squared test, p=0.6) (*Table 1*). When stratifying the general population from the Society Islands by age (≤16 years and >16 years), there was a decline in adults in the two consecutive cross-sectional studies conducted in 2014 and 2015, from 35.4% (22.2–50.5%) to 21.3% (18.2–24.5%) (*Figure 1*). A decline in adults was still observed, albeit with larger uncertainty, when the two datasets were standardised according to the age distribution of the population, with age-adjusted seroprevalence decreasing from 32.0% (16.7–62.1%) to 26.0% (20.1–33.9%) (*Table 2*).

In Fiji, in the subset of individuals who were aged over 16 years (*n* = 122), there was a decrease in seroprevalence by MIA from 24% (17–33%) in 2015 to 7.3% (3.4–13%) 2017 (*Figure 1*). There were two seroconversions in the collected samples over this period but 23 seroreversions (McNemar's test, p<0.0001) (*Table 3*). In contrast seroprevalence in participants aged 16 and under (*n* = 67) remained relatively stable over this period (*Figure 1*), with four seroconversions and five seroreversions (McNemar's test, p=1) (*Table 3*).

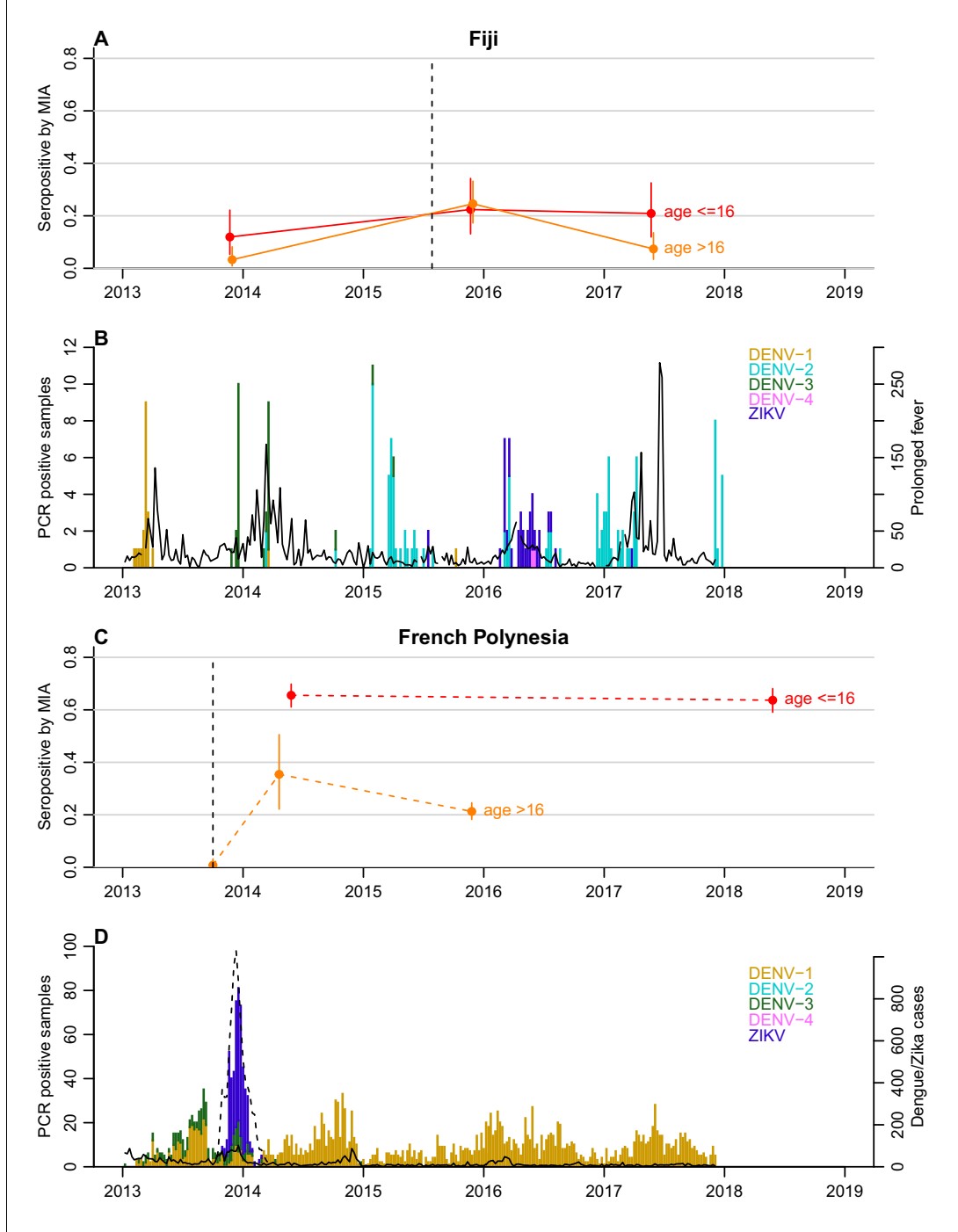

**Figure 1.** Dynamics of ZIKV seroprevalence following outbreaks in Fiji and French Polynesia. (**A**) Seroprevalence by MIA in Fiji. Red, seroprevalence and 95% confidence intervals for children (aged ≤16 years). Orange, seroprevalence and 95% confidence intervals for adults (aged >16 years). Solid lines, trends in data collected from the same individuals. Dotted line indicates the first confirmed ZIKV case. (**B**) Epidemiological dynamics in Fiji between 2013 and 2018. Coloured bars show number of PCR-confirmed samples of different DENV serotypes and ZIKV in Fiji; black lines show reported prolonged fever in Fiji from the Pacific Syndromic Surveillance System (*World Health Organization, 2019*). There was a major outbreak of DENV-3 outbreak in 2013–14 (*Kucharski et al., 2018a*) with a smaller DENV-2 outbreak in early 2017 (*Aubry et al., 2019*). (**C**) Seroprevalence by MIA in French Polynesia. Dashed lines, trends in seroprevalence between population representative cross-sectional surveys. Note that the pre-outbreak samples were collected between July 2011 and October 2013; for brevity, the latest possible collection date is used in the plot. (**D**) Epidemiological dynamics in French Polynesia between 2013 and 2018. Solid black line shows reported symptomatic dengue cases; dashed lines showed reported symptomatic Zika cases. In French Polynesia, between the sampling periods, there were no reported DENV outbreaks for serotypes 2,3,4, and there was hyper-endemic DENV-1 circulation. In April 2019, a DENV-2 outbreak was declared, the first since 1997 (*Aubry et al., 2019*).

*Figure 1 continued on next page*

*Figure 1 continued*

The online version of this article includes the following figure supplement(s) for figure 1:

**Figure supplement 1.** Seroprevalence against DENV-1 in Fiji and French Polynesia, by age group.
**Figure supplement 2.** Seroprevalence against DENV-2 in Fiji and French Polynesia, by age group.
**Figure supplement 3.** Seroprevalence against DENV-3 in Fiji and French Polynesia, by age group.
**Figure supplement 4.** Seroprevalence against DENV-4 in Fiji and French Polynesia, by age group.

In order to assess whether the decline in ZIKV seroprevalence was also observed for other circulating *Flaviviruses*, the MIA seroprevalence pattern against each of the four DENV serotypes was analyzed in both countries, by age group (*Figure 1—figure supplements 1–4*). In Fiji, seroprevalence for DENV-1, DENV-2 and DENV-4 increased in participants in both age groups between 2013 and 2017 (*Figure 1—figure supplements 1*, *2* and *4*). DENV-3 seroprevalence also increased in both age groups between 2013 and 2015 following an outbreak in 2013–14 (*Kucharski et al., 2018a*) and then declined in 2017 from 44% (32–57%) to 40% (28–52%) in children (McNemar's test, p=0.6) and from 59% (50–68%) to 49% (40–58%) in adults (McNemar's test, p=0.01) (*Figure 1—figure supplement 3*). In French Polynesia between 2014 and 2018, seroprevalence in children aged under 16 years showed no evidence of a change for DENV-1 and DENV-2 (chi-squared test, p=0.1917 and p=1, respectively) (*Figure 1—figure supplements 1–2*) and decreased for DENV-3 and DENV-4 (chi-squared test, p<0.0001 and p=0.0085, respectively) (*Figure 1—figure supplements 3–4*). In adult participants from the general population, seroprevalence for all four DENV serotypes declined between 2014 and 2015.

The age-adjusted values for seroprevalence by MIA for the four DENV serotypes were similar to the raw values (*Table 2*), suggesting that the decline in French Polynesia could not be explained by differences in sampling by age. However, a higher proportion of the samples in 2014 tested positive by MIA for all four DENV serotypes (*Table 4*), suggesting that the sampling included a group at higher risk for arbovirus infection than those sampled in 2015. To check that the estimated decline in ZIKV seroprevalence was not an artefact of this sampling bias, we re-estimated seroprevalence for the four DENV serotypes and ZIKV using a bootstrap sample of the 2014 responses, with replacement, weighted by the DENV exposure profile (excluding the virus of interest) in the 2015 survey so that the bootstrap sample of the 2014 responses had a similar DENV exposure profile as in the 2015 responses. For example, when generating bootstrap estimates for DENV-1 in 2014, we resampled participants based on the distribution of number of exposures to DENV-2, DENV-3, and DENV-4 in the 2015 data (*Table 5*). After adjusting for prior exposure, there was no significant decline in seroprevalence for DENV-1, DENV-3, or DENV-4, which had all circulated in the five years preceding the 2014 data collection, whereas the decline in ZIKV was still present (chi-squared test, p=0.0047).

To explore dynamics of antibody waning at the individual level, we performed neutralization assays (NT) on a subset of 45 participants from Fiji for whom sufficient sera were available to test against ZIKV from all three collection periods, focusing on those who were seropositive to ZIKV by MIA in 2013 or 2015. We found that in the 31 individuals who were ZIKV seronegative by NT (i.e. log titre <2) in 2013 and had a rise in log titre ≥2 against ZIKV between 2013 and 2015, anti-ZIKV antibody responses waned significantly in 2017, with an average decline in log titre of −1.94 (t-test, p<0.0001) (*Figure 2A* and *Table 6*). In total, four participants seroreverted between 2015 and 2017;

**Table 2.** Age-adjusted seroprevalence by MIA in participants aged over 16 in the general population of the Society Islands in French Polynesia, based on serosurveys conducted in 2014 (*n* = 48) and 2015 (*n* = 672).

| Virus | 2014 seroprevalence (95% CI) | 2014 age-adjusted seroprevalence (95% CI) | 2015 seroprevalence (95% CI) | 2015 age-adjusted (95% CI) |
|---|---|---|---|---|
| DENV1 | 85 (72–94) | 83 (55–100) | 80 (77–83) | 80 (71–91) |
| DENV2 | 48 (33–62) | 50 (28–87) | 19 (16–22) | 21 (15–21) |
| DENV3 | 75 (60–86) | 72 (47–100) | 56 (52–60) | 55 (48–64) |
| DENV4 | 63 (47–76) | 65 (40–100) | 42 (38–46) | 45 (38–54) |
| ZIKV | 35 (22–50) | 32 (16–62) | 21 (18–25) | 26 (20–34) |

[*]chi-squared test comparing 2014 bootstrap estimates with 2015 results.

**Table 3.** Detection of IgG by MIA against ZIKV in the paired samples from participants aged under and over 16 years recruited during October-November 2015 and June 2017 in the Central division in Fiji (n = 189).

Age groups are defined using age of participants when recruited to the study in 2013.

| | 2017 | | | | | |
|---|---|---|---|---|---|---|
| | ≤16 years | | >16 years | | Total participants | |
| **2015** | **ZIKV+** | **ZIKV-** | **ZIKV+** | **ZIKV-** | **ZIKV+** | **ZIKV-** |
| ≤16 years | | | | | | |
| ZIKV+ | 10 | 5 | – | – | – | – |
| ZIKV- | 4 | 48 | – | – | – | – |
| >16 years | | | | | | |
| ZIKV+ | – | – | 7 | 23 | – | – |
| ZIKV- | – | – | 2 | 90 | – | – |
| Total Participants | | | | | | |
| ZIKV+ | – | – | – | – | 17 | 28 |
| ZIKV- | – | – | – | – | 6 | 138 |

all had a log titre of 4 against ZIKV in 2015. We observed a similar effect when we analysed all participants who had a rise in log titre of at least 2 between 2013–15, regardless of serostatus in 2013 (*Figure 2—figure supplement 1*).

To test whether the dynamics of anti-ZIKV antibody waning were different from the responses to DENV infection, we compared results for ZIKV to the neutralization response following a DENV-3 infection in the same cohort from Fiji. There was a large DENV-3 epidemic during 2013–14 in Fiji (*Osuna et al., 2016*), which meant most seroconversions to DENV-3 occurred between the collection of samples in 2013 and 2015. In those individuals that seroconverted to DENV-3 (*n* = 19) or ZIKV (*n* = 31) between 2013 and 2015, the initial rise in NT log titres against ZIKV was larger than for DENV-3, with a mean change of 5.0 and 3.37 respectively (*Figure 2B* and *Table 6*). All individuals who had seroconverted to DENV-3 remained seropositive to the virus in 2017, while four individuals who had seroconverted to ZIKV were seronegative in 2017. Although the NT log titres increased by a mean of 0.89 for DENV-3 between 2015 and 2017 (two-sided t-test, p=0.04), log titres against ZIKV declined by a mean of 1.94 over the same period (two-sided t-test, p<0.001) (*Figure 2A* and *Table 6*).

In Fiji, there was a delay of around 18 months between the end of the 2013–14 DENV-3 epidemic and collection of samples in 2015. As DENV titres can wane following infection, particularly in individuals with a prior DENV exposure (*Clapham et al., 2016*), titres against DENV-3 in Fiji may

**Table 4.** Age distribution and profile of DENV exposure history in two cross-sectional surveys conducted in the general population from the Society Islands, French Polynesia, in 2014 and 2015.

While the age distribution is similar in both studies, the sample in 2014 has a higher proportion of individuals who have tested positive for infection from all four DENV serotypes by MIA.

| Variable | 2014 (*n* = 49) | 2015 (*n* = 700) |
|---|---|---|
| Age distribution (*median* [*IQR*]) | 47 [29-56] | 43 [29-57] |
| Number of DENV serotypes positive at time of sample collection (*n* [%]) | | |
| 0 | 3 [0.061] | 118 [0.17] |
| 1 | 6 [0.12] | 163 [0.23] |
| 2 | 11 [0.22] | 159 [0.23] |
| 3 | 11 [0.22] | 154 [0.22] |
| 4 | 18 [0.37] | 106 [0.15] |

**Table 5.** Bootstrap estimated seroprevalence for each of the four DENV serotypes and ZIKV adjusted for sampling bias in two cross-sectional surveys conducted in the general population from the Society Islands, French Polynesia, in 2014 and 2015.

Results from the cross-sectional surveys in the Society Islands, French Polynesia, in 2014 and 2015 show a decline in seroprevalence by MIA against all 4 DENV serotypes and ZIKV. However, the 2014 sample included more individuals that tested positive for >1 DENV serotype and are assumed to be a higher risk group. We used a bootstrap method with 10,000 iterations which estimated seroprevalence from a sample of the 2014 dataset, taken with replacement, weighted by the exposure distribution to other DENV viruses in the 2015 survey. After adjusting for the sample bias, there was no evidence of a decline in seroprevalence for DENV-1, DENV-3, or DENV-4, which had circulated in the years preceding the 2014 sample collection (**World Health Organisation, 2019**), but there remained strong evidence that ZIKV seroprevalence declined between 2014–15.

| Virus | 2014 seroprevalence (95% CI) (n = 49) | 2014 bootstrap estimates of seroprevalence (95% CI) | 2015 seroprevalence (95% CI) (n = 700) | *p*-value[*] |
|---|---|---|---|---|
| DENV1 | 86 (73–94) | 74 (61–86) | 80 (77–83) | 0.36 |
| DENV2 | 47 (33–62) | 38 (24–53) | 18 (15–21) | 0.0008 |
| DENV3 | 76 (61–87) | 64 (51–78) | 55 (51–59) | 0.21 |
| DENV4 | 63 (48–77) | 50 (37–65) | 42 (38–46) | 0.42 |
| ZIKV | 37 (23–52) | 42 (29–55) | 22 (19–25) | 0.0047 |

[*]chi-squared test comparing 2014 bootstrap estimates with 2015 results.

therefore have had more time to wane and reach a stable persistent level than titres against ZIKV, which may have circulated later than DENV-3. We therefore analysed changes in titre for participants who were initially seronegative to DENV-1 and DENV-2, which were circulating at low levels in Fiji between the two serological surveys in 2013 and 2015 (*Figure 1*). As with DENV-3, we found no evidence of a subsequent overall decline during 2015–17 for those participants who seroconverted to DENV-1 or DENV-2 during 2013–15 (*Figure 2—figure supplement 2*).

Of the 45 participants tested by neutralization assay, nine were initially seropositive to ZIKV by NT in 2013. Fitting a generalized additive model to these data, we found that higher baseline mean NT log titres against DENV were associated with an increased probability of seropositivity to ZIKV (*Figure 3A*). In contrast, higher baseline mean DENV titres were not associated with increased seropositivity by MIA in 2013. There was little difference between the assay results in the 2015 samples (*Figure 3B*), but we did find evidence of a difference in the 2017 results, with 15/45 participants positive by MIA and 31/45 positive by NT. This difference was associated with participants' 2013 DENV titres: those with intermediate DENV titres in 2013 had a significantly lower probability of being seropositive in the MIA in 2017 compared to NT (*Figure 3C*).

## Discussion

Analyzing data from serological surveys conducted in French Polynesia and Fiji at different time points after the first reported autochthonous ZIKV transmission, we found evidence of a decline in ZIKV seroprevalence. The high number of participants from the Fijian cohort that seroreverted between 2015 and 2017 suggested that anti-ZIKV antibody levels waned in these individuals to the point that they were no longer detectable by MIA. Using a neutralization assay to test longitudinal sera collected in Fiji, we found that the mean change in neutralizing antibody titres against ZIKV also decreased significantly between 2015 and 2017, showing that individual-level antibody titres against ZIKV as well as overall seroprevalence decreased over time. In contrast, over the same period, neutralizing antibody titres against DENV-3, a closely related *Flavivirus* which caused a large epidemic in Fiji in 2013–2014 (*Kucharski et al., 2018a*), remained stable.

In both countries we found seroprevalence against ZIKV in individuals aged over 16 declined over the two-year period following an outbreak, while the overall level of seroprevalence persisted in children. This pattern was unique to ZIKV compared to DENV in both countries. It is possible that this is related to the DENV immunological profile of individuals, given that the older population is likely to have experienced more DENV infections over their lifetime. If an individual has experienced prior DENV infections, high numbers of weakly neutralizing cross-reactive B cells may outcompete naïve B cells for ZIKV antigen (*Midgley et al., 2011*), leading to a short-term boost in antibody response

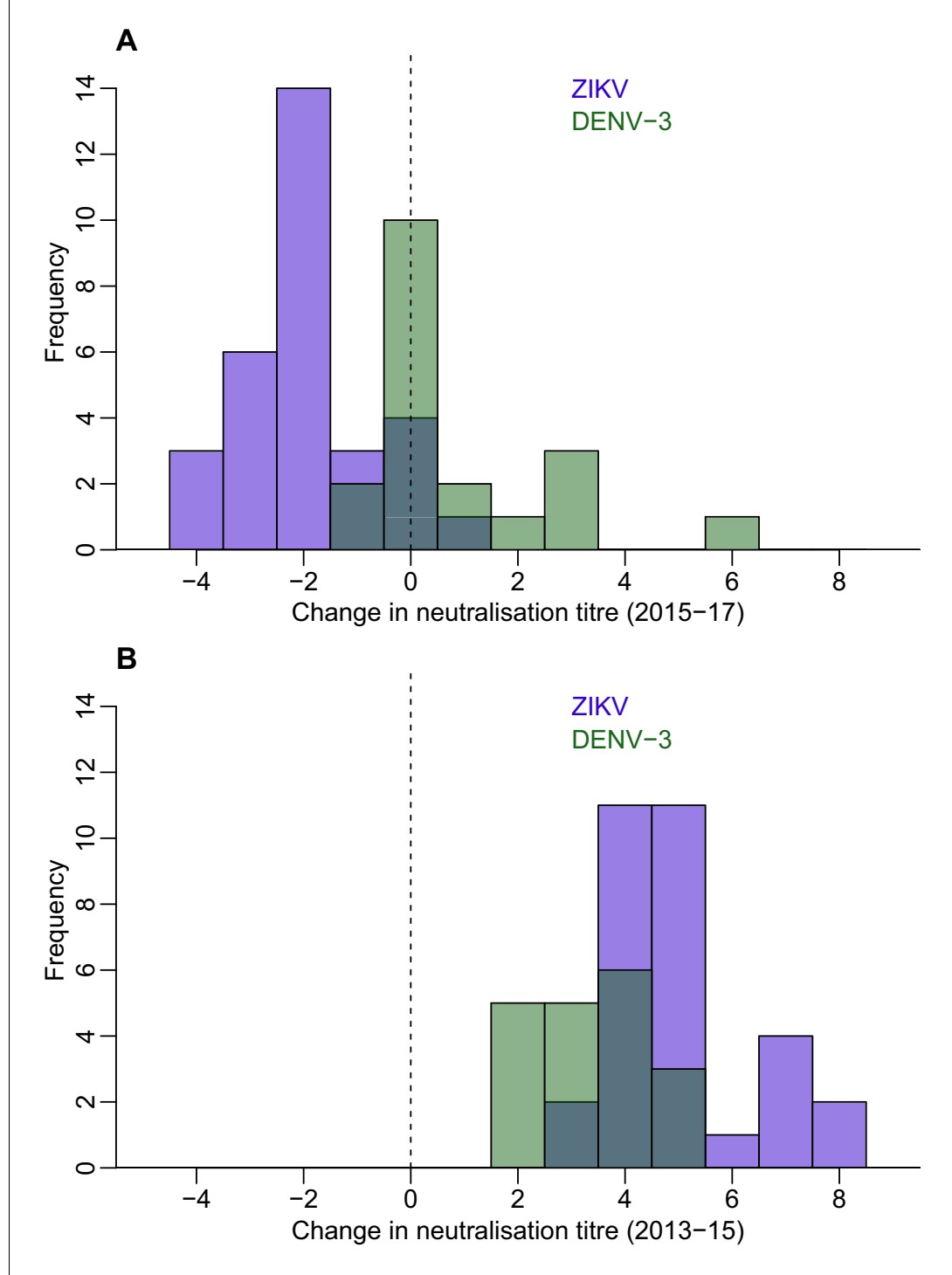

**Figure 2.** Waning of neutralizing antibody responses against ZIKV and DENV-3 in Fiji for participants who were seronegative to each virus in 2013 and seroconverted in 2015. (**A**) Histogram of change in neutralization assay log titre against DENV-3 ($n = 19$) and ZIKV ($n = 31$) between 2015–2017 for individuals who seroconverted to these respective viruses between 2013–2015 (i.e. log titre < 2 in 2013 and log titre ≥ 2 in 2015). (**B**) Histogram of change in log titre against DENV-3 and ZIKV for individuals who seroconverted to these respective viruses during 2013–2015.
The online version of this article includes the following figure supplement(s) for figure 2:

**Figure supplement 1.** Waning of neutralizing antibody responses against ZIKV and DENV-3 in Fiji for participants who had a four-fold rise between 2013 and 2015.

*Figure 2 continued on next page*

*Figure 2 continued*

**Figure supplement 2.** Waning of neutralizing antibody responses against DENV-1 and DENV-2 in Fiji for participants who were seronegative in 2013 and seroconverted in 2015.

against ZIKV following ZIKV infection (*Robbiani et al., 2017*) but not a persistent specific response; a similar phenomenon has been observed for other antigenically variable viruses like influenza (*Kucharski et al., 2018b*). In the 2017 samples, more participants remained seropositive in the neutralization assay – which measures the overall ability of sera to neutralize ZIKV – than in the MIA, which tests for IgG antibodies against domain III of the envelope glycoprotein. This difference was greatest for participants who had intermediate baseline titres to DENV in 2013 (*Figure 3C*), which would support the hypothesis that prior DENV exposure may result in a detectable short-term specific response against ZIKV following ZIKV infection (as measured by MIA), but not a persistent specific response.

To our knowledge, the only other study to date that has investigated the long-term persistence of neutralizing antibodies against ZIKV was conducted in 62 residents of Miami (Florida, USA), who had a confirmed ZIKV infection in 2016 (*Griffin et al., 2019*). This cross-sectional study found that all participants had neutralizing antibodies against ZIKV 12–19 months after infection. This study also found that at least 37% of the participants had no evidence of past DENV infection, which is consistent with the hypothesis that anti-ZIKV immune responses may persist longer in populations that have had less exposure to DENV. More data are therefore needed to test hypotheses about the potential impact of pre-existing anti-DENV immune response on anti-ZIKV antibody waning.

Although we found evidence of a decline in seroprevalence for antibodies against domain III of the envelope glycoprotein, as well as waning neutralizing antibody responses following two ZIKV outbreaks, the implications for susceptibility to future ZIKV infection remain unclear. Given the antigenic similarity of DENV and ZIKV (*Priyamvada et al., 2016*), it is commonly assumed that the immune response to ZIKV infection will be similar to that following DENV infection. High levels of neutralizing antibodies to DENV have been shown to correlate with protection from symptomatic infection (*Katzelnick et al., 2016*). Moreover, infection with a single DENV serotype can confer life-long immunity to the infecting serotype as well as a transient period of cross-neutralization against heterologous serotypes (*Wahala and Silva, 2011*). However, it is unclear in the context of ZIKV what the relationship is between a specific titre value and susceptibility to further infection. A key aim for future work will be to establish how waning antibody levels as measured by MIA and neutralization assays may impact protective immunity, and hence susceptibility to reinfection in populations that have already experienced transmission of ZIKV.

There are some additional limitations to our analysis. First, we did not have reverse transcription polymerase chain reaction (RT-PCR) confirmation of ZIKV infection in individuals sampled in this study. We have presented analysis of representative serological surveys in two locations with known, RT-PCR-confirmed ZIKV outbreaks (*Mallet et al., 2015*; *World Health Organisation, 2015*). However, RT-PCR confirmation for ZIKV at the individual level remains difficult to obtain, in particular

**Table 6.** Change in neutralization titre between 2013–2017 in a cohort of 45 study participants in Fiji.

ZIKV and DENV-3 both circulated in Fiji between the collection of samples in 2013 and 2015, with ZIKV first reported in July 2015 and DENV-3 circulating between October 2013 and January 2015. Neutralization titre levels rose significantly over this period. Between 2015 and 2017, DENV-3 titre levels still increased with a mean change in tire of 0.89. By contrast, the mean change in ZIKV titre over this period decreased (−1.9).

| Virus | 2013–2015 change, Mean [95% CI] | p-value[*] | 2015–2017 change, Mean [95% CI] | p-value[*] |
|---|---|---|---|---|
| ZIKV (n=31) | 5 [4.5, 5.5] | <0.0001 | −1.9 [-2.4,–1.5] | <0.0001 |
| DENV3 (n=19) | 3.4 [2.9, 3.9] | | 0.89 [0.046, 1.7] | |

[*] t-test comparing change in neutralization titre for ZIKV and DENV-3 between 2013–2015, and 2015–2017.

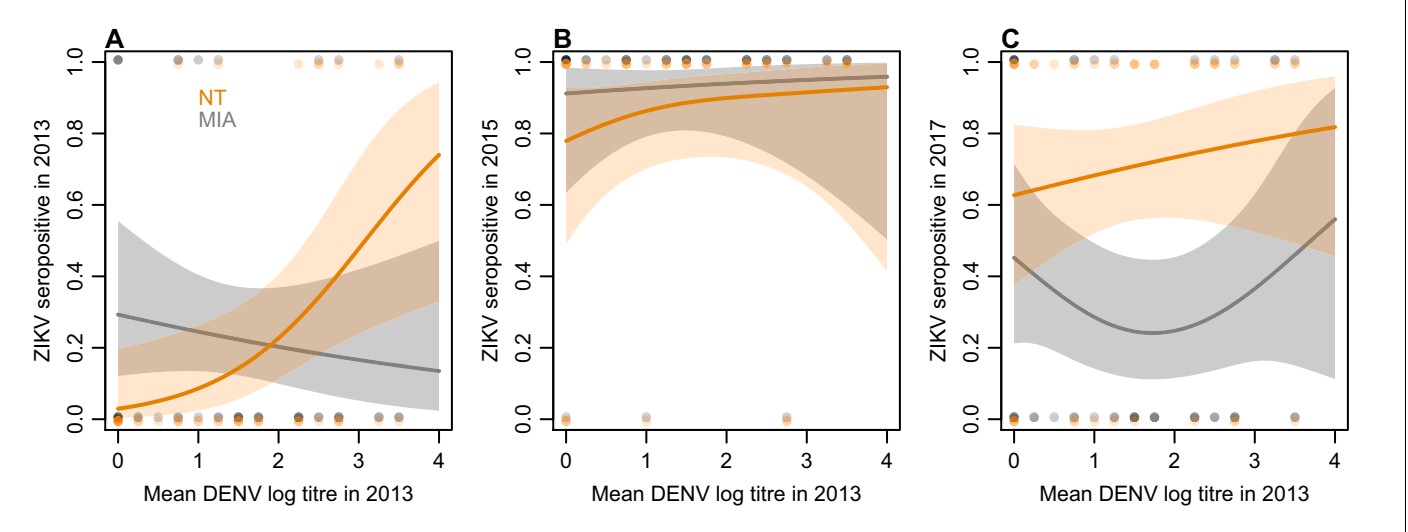

**Figure 3.** Relationship between mean DENV log neutralization titre across the four serotypes in 2013 and ZIKV seroprevalence using different assays, in a subset of 45 participants. (**A**) Seroprevalence by MIA, shown in grey, and neutralization test (NT), shown in orange, for sera collected in 2013. Line shows prediction from GAM fitted to each dataset, with shaded region showing 95% CI, and points show raw data. (**B**) Seroprevalence for sera collected from the same participants in 2015. (**C**) Seroprevalence for sera collected from the same participants in 2017.

from blood samples, and there have been relatively few confirmations globally compared to the number of suspected cases (*Ferguson et al., 2016*), let alone analysis of long-term antibody dynamics in RT-PCR confirmed patients. In French Polynesia, there were approximately 32,000 reported clinical cases of ZIKV infection, but only 297 documented RT-PCR-confirmed cases (*Mallet et al., 2015*). As a result, antibody responses in RT-PCR-confirmed cases may not necessarily be representative of immune responses against ZIKV in the wider population, particularly following asymptomatic infection. Although MIA seropositivity in our study was defined using control sera collected over a year after RT-PCR-confirmed infection, our results suggest that this threshold may not detect long-term waning responses in individuals who had unreported, and likely less severe, infections.

Our analysis was also limited by study design. In French Polynesia, surveys were cross-sectional, so we were unable to examine temporal antibody dynamics at the individual level. However, both cross-sectional studies of the general population were conducted using population representative cluster sampling (*Aubry et al., 2017*) in the same remote island locations with stable population composition, which enabled robust comparisons of overall seroprevalence. We did identify one potential source of sampling bias with different DENV exposure profiles in the two surveys, but our conclusions of declining seroprevalence for ZIKV persisted once we adjusted for this bias. We also used a different serological testing method between the studies in French Polynesia in 2014 and 2015. However, both used the same recombinant antigens and it has been shown that there was good agreement between ELISA and MIA in the 2014 samples (see Materials and methods). In Fiji, a strength of our study was the collection of longitudinal samples from the same individuals at three time points. However, our sample size was limited given the logistical challenge of recontacting participants twice over a four-year period. These data provided strong evidence that ZIKV seroprevalence declined over the two-year period following first reports of circulation, but our sample size was insufficient to fully explore the potential effect of anti-DENV pre-existing immunity on anti-ZIKV antibody waning once we stratified individuals by previous DENV exposure. Although the outbreaks of DENV-3 in Fiji and ZIKV in French Polynesia were well-documented and occurred over a relatively brief period of time (*Figure 1*), it was harder to identify the likely time of infection for other viruses – such as ZIKV in Fiji or DENV in French Polynesia – in our study populations. Several participants in Fiji were seropositive to ZIKV by neutralization assay (NT) in 2013, but this result may be influenced by cross-reaction; participants who had high pre-existing titres to DENV in 2013 were more likely to be seropositive by NT (*Figure 3A*). In our main analysis of titre dynamics, we therefore focused on the subset of participants who were seronegative by NT in 2013 (*Figure 2*). However, we obtained

the same conclusion when participants who were initially seropositive were also considered (*Figure 2—figure supplement 1*).

The global ZIKV epidemic began in the Pacific islands in 2013 before spreading in Central and South America from 2015. Seroprevalence studies following ZIKV epidemics in Latin America have been reported but data have either been non-representative (*Netto et al., 2017*) or not enough time had elapsed since the outbreak to observe long-term dynamics (*Rodriguez-Barraquer et al., 2019*; *Zambrana et al., 2018*). To our knowledge, these are the first studies of community seroprevalence over a long-term period following a ZIKV outbreak. Therefore, patterns observed in Pacific islands may be an early indication of what might happen to seroprevalence in Latin America where ZIKV outbreaks began two to three years after the French Polynesia epidemic (*Cao-Lormeau et al., 2014*; *Bogoch et al., 2016*).

In the short-term, our findings have implications for the design of follow up studies of ZIKV. Our results provide evidence that levels of seroprevalence one to two years following ZIKV circulation may be lower than previously expected and study designs may need to be adapted to reflect this, particularly in settings that exhibit long-term low level circulation of ZIKV as opposed to large sporadic outbreaks (*Ruchusatsawat et al., 2019*). For example, estimates of microcephaly risk may be inflated if derived from long-term seroprevalence data that underestimate the true extent of infection within the population, and results of clinical trials could also be biased if post-outbreak seroprevalence is used an indicator of infection within a population (*Cohen, 2018*). In the longer-term, our results demonstrate the value of longitudinal serological studies of flaviviruses, and analysis using multiple serological tests, including neutralization assays (*Clapham et al., 2016*). Such studies will be essential to understand different aspects of the short and long-term immune antibody response against ZIKV, and how prior exposures to DENV may influence these responses.

## Materials and methods

### Study location and participants

#### French polynesia

Four separate ZIKV serosurveys were previously conducted in the Society Islands (*Table 1*). As reported previously (*Aubry et al., 2017*; *Aubry et al., 2015a*), a first serosurvey ($n$ = 593) was conducted in adult blood donors recruited between July 2011 and October 2013, before the ZIKV outbreak that occurred between October 2013 and April 2014 (*Cao-Lormeau et al., 2014*). Two population-representative serosurveys were conducted among the general population, firstly between February and March 2014 ($n$ = 196), and then between September and November 2015 ($n$ = 700). The two studies in the general population both spanned a range of adult age groups (*Table 7*). An additional serosurvey was conducted among schoolchildren between May and June 2014 ($n$ = 476). Finally, a fifth serosurvey was conducted among schoolchildren in the Society Islands in June 2018 ($n$ = 457) using the same protocol as in 2014 (*Aubry et al., 2017*).

#### Fiji

Three serosurveys were conducted in Fiji (*Table 1*). Individuals were first recruited into a population-representative community-based typhoid/leptospirosis seroprevalence study between September and November 2013 (*Watson et al., 2017*) ($n$ = 1,787), before autochthonous transmission of ZIKV was first detected in July 2015 (*World Health Organisation, 2015*). Briefly, nursing zones were randomly selected, from which one individual from 25 households in a randomly selected community was recruited. Participants who had consented to being contacted again for health research were subsequently recruited in November 2015 in 23 communities in Central Division through last known addresses, phone numbers and the assistance of local nurses ($n$ = 327) (*Kama et al., 2019*). A third follow-up serosurvey was conducted in June 2017 using the same protocol as in 2015 ($n$ = 321) (*Kucharski et al., 2018a*). Follow-up surveys were only performed in Central Division, which was the focus of a DENV-3 outbreak in 2013–14 (*Kucharski et al., 2018a*). Only blood samples serially collected from the same participants ($n$ = 189) in 2013, 2015 and 2017 were analyzed in the main results presented in this study.

**Table 7.** Age distribution of study population in French Polynesia.
Overall population distribution shown, along with total samples collected in each age group in 2014 and 2015 serosurveys.

| Age range | Population estimate (2017) | Samples in 2014 study | Samples in 2015 study |
|---|---|---|---|
| 0–9 | 42,770 | 0 | 0 |
| 10–19 | 43,705 | 3 | 22 |
| 20–29 | 48,914 | 10 | 135 |
| 30–39 | 42,144 | 5 | 131 |
| 40–49 | 40,886 | 8 | 119 |
| 50–59 | 34,478 | 15 | 128 |
| 60–69 | 21,099 | 2 | 85 |
| 70–79 | 10,481 | 5 | 46 |
| 80–89 | 3773 | 0 | 9 |
| 90+ | 416 | 0 | 0 |

## Informed consent and ethics approvals
### French polynesia
The five serosurveys were approved by the Ethics Committee of French Polynesia (ref 61/CEPF 08/29/2013, 60/CEPF 06/27/2013, 74/CEPF 05/04/2018, and 75/CEPF 05/04/2018).

### Fiji
The original 2013 study, and the 2015 and 2017 follow up studies were approved by the Fiji National Research Ethics Review Committee (ref 2013–03, 2015.111.C.D, 2017.20.MC) and the London School of Hygiene and Tropical Medicine Observational Research Ethics Committee (ref 6344, 10207, 12037).

## Serological analysis
### French polynesia
Serum samples collected from blood donors between July 2011 and October 2013 and samples collected from the general population and schoolchildren in 2014 were all tested for presence of IgG antibodies against ZIKV and each of the four DENV serotypes using a recombinant antigen-based indirect ELISA as reported previously (*Aubry et al., 2017*; *Aubry et al., 2015a*). Samples collected from the general population in 2015 and from schoolchildren in 2018 were tested by microsphere immunoassay (MIA) using the same recombinant antigens as for the ELISA (*Cao-Lormeau et al., 2016*; *Aubry et al., 2017*; *Kama et al., 2019*). Recombinant antigens used in both assays comprised domain III of the envelope glycoprotein of ZIKV, DENV-1, DENV-2, DENV-3, or DENV-4 strains (respective GenBank accession no. KJ776791, AF226686.1, FM986654, FJ44740.1, FM986672.1) and were produced using the *Drosophila* S2 expression system (Life Technologies, USA) as previously detailed (*Aubry et al., 2015b*). Serostatus was defined by a cut-off determined using positive and negative control sera analyzed by ROC curve. The sensitivity and specificity of the MIA assay were respectively 100% and 100% for DENV-1, 89.5% and 97.1% for DENV-2, 100% and 100% for DENV-3, 96.9% and 100% for DENV-4, and 79.6% and 94.9% for ZIKV. The positive control sera for ZIKV was collected 13 months after RT-PCR confirmed infection. In the serosurvey conducted among the general population of the five archipelagos in French Polynesia in 2014 (*Aubry et al., 2017*), 196 samples were tested using both ELISA and MIA: among the 97 serum samples that tested positive for anti-ZIKV IgG by ELISA, 78 (80%) were also found positive by MIA; and among the 99 serum samples that tested negative for anti-ZIKV IgG by ELISA, 70 (71%) were also found negative by MIA. This produced a value of Cohen's κ = 0.51 (*Aubry et al., 2017*).

## Fiji

All serum samples collected in Fiji were tested using MIA to detect IgG antibodies against ZIKV and all four DENV serotypes as previously reported (*Cao-Lormeau et al., 2016*; *Aubry et al., 2017*; *Kucharski et al., 2018a*). To follow the evolution of antibody titres at the individual level, a subset of samples collected from the same individuals in 2013, 2015 and 2017 were tested for the presence of neutralizing antibodies against ZIKV and each of the four DENV serotypes using a neutralization assay as previously described (*Cao-Lormeau et al., 2016*). This subset of samples was selected to include all participants who were seropositive to ZIKV by MIA in 2013 and 2015, as well as one participant just below the seropositivity threshold, and for whom we had sufficient longitudinal serum available from 2013, 2015 and 2017 to test by neutralization assay ($n = 45$). We also tested samples from an additional 24 participants from the same cohort who were seropositive to ZIKV by MIA in 2013 or 2015 and for whom we had sufficient serum from 2013 and 2015 to test by neutralization assay, but no matched sample from the 2017 follow up survey (i.e. 69 paired samples in total). ZIKV log titres in the neutralization assay followed a bimodal distribution, which supported the use of a log titre of $\geq 2$ as a cutoff for seropositivity (*Figure 4*).

Of the 9/45 participants with three samples who were seropositive to ZIKV by neutralization assay in 2013, all were seropositive to at least one DENV serotype (*Figure 4—figure supplement 1*). To assess the potential for cross-reactive antibody responses, we examined the correlation between changes in log titre to different viruses between 2013 and 2015. Among the 20/69 paired samples that tested seronegative against all five viruses in 2013 and were re-tested in 2015, there was no evidence of an association between changes in ZIKV titre and changes in titre against any of the DENV serotypes, suggesting that the changes in ZIKV titre were unlikely to be strongly influenced by DENV cross-reaction (*Figure 4—figure supplement 2*). However, the 49/69 participants who had a pre-2013 DENV exposure and a large rise against ZIKV between 2013–15 tended to exhibit a smaller rise against DENV viruses (*Figure 4—figure supplement 3*).

A previous study, which tested serological samples from Fiji across three divisions (*Kama et al., 2019*), found that of the samples reactive by MIA, 66/83 (79.5%) exhibited neutralizing activity for ZIKV ($\kappa = 0.71$) and 109/112 (97.3%) for DENV ($\kappa = 0.80$). In this study, we tested what proportion of samples for the 45 participants in the full dataset (i.e. 135 samples in total) that were seropositive or seronegative by MIA had the same result by the neutralization assay. We found that 54/68 (79.4%)

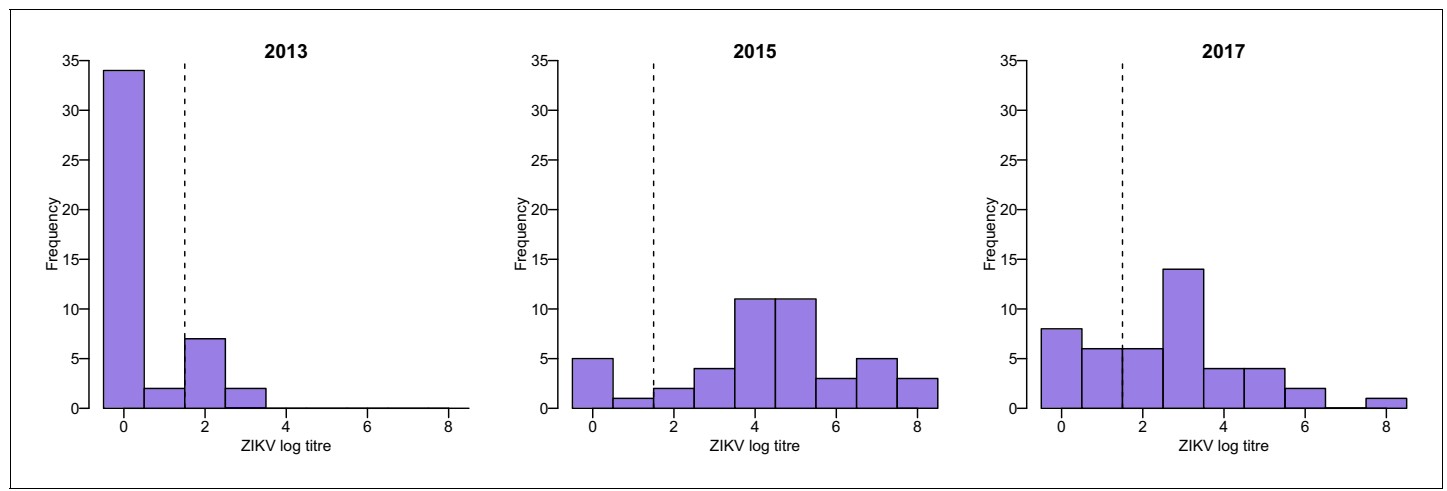

**Figure 4.** Distribution of ZIKV neutralization titres in the Fiji serosurveys. Results shown for 45 participants who had samples available from 2013, 2015, and 2017. Dashed line shows the threshold used to define seropositivity.
The online version of this article includes the following figure supplement(s) for figure 4:

**Figure supplement 1.** Individual-level neutralization log titres against the four DENV serotypes and ZIKV in Fiji.
**Figure supplement 2.** Correlation between rise in DENV and ZIKV neutralization log titres between 2013–2015 for participants who were initially seronegative (i.e. log titre <2) to all five viruses in 2013 ($n = 20$).
**Figure supplement 3.** Correlation between change in DENV and ZIKV log titres between 2013–2015 for participants who were initially seropositive (i. e. log titre $\geq 2$) to at least one DENV virus in 2013 ($n = 49$).

samples that were positive to ZIKV by MIA were also positive in the neutralization assay, and 42/67 (62.7%) who were seronegative were also negative by neutralization assay (κ = 0.42). We also calculated the proportion positive by neutralization assay that had the same result by MIA. We found that 54/79 (68.4%) samples that were positive to ZIKV in the neutralization assay were also positive by MIA, and 42/56 (75%) who were seronegative were also negative by MIA.

## Statistical analysis

For data from Fiji, where serial samples were collected from the same individual, changes in seroprevalence between studies were tested using McNemar's test. In French Polynesia, chi-squared tests were performed to test for evidence of a change in seroprevalence between two cross-sectional surveys. Changes in mean log titre between groups were analyzed using a t-test. To analyse the potentially non-linear relationship between DENV neutralization titres and seroprevalence by MIA and neutralization test (*Figure 3*), we used a generalized additive model via the mcgv package in R (*Wood, 2019*). The model was of the form $g(E(y))=b + f(x)$, where *y* was the binary outcome variable, *x* was the predictor (i.e. titre), *g* was the link function, *b* was the intercept, and *f* was a smooth function represented by a penalized regression spline. Mean DENV titre was calculated as the mean of log titres against the four DENV serotypes for each participant. All data and code used in the analysis are available at: https://github.com/a-henderson91/zika-sero-pacific/ (*Henderson and Kucharski, 2019*; copy archived at https://github.com/elifesciences-publications/zika-sero-pacific/settings).

## Acknowledgements

We are grateful to the Minister for Education of French Polynesia and to the directors, teachers, nurses and schoolchildren from the elementary and junior high schools selected for the serosurvey in 2018. We greatly thank all the participants and community leaders in Fiji who generously contributed to the study over the three visits. We would like to acknowledge the work of the field teams: Jessica Paka, Amele Ratevono, Warren Fong, Manisha Prakash, Jonetani Bola, Mosese Ligani, and Taina Naivalu (2017); Meredani Taufa, Adi Kuini Kadi, Jokaveti Vubaya, Colin Michel, Mereani Koroi, Atu Vesikula, and Josateki Raibevu (2015); Dr. Kitione Rawalai, Jeremaia Coriakula, Ilai Koro, Sala Ratulevu, Ala Salesi, Meredani Taufa, and Leone Vunileba (2013). We thank Rina Kumar, Sokoveti Covea, Taina Naivalu and Vinaisi Duituturaga for their assistance with sample preparation. We would also like to thank Eric J Nilles of the World Health Organization Western Pacific Region.

## Additional information

### Funding

| Funder | Grant reference number | Author |
|---|---|---|
| Ministry for Europe and Foreign Affairs | Pacific funds 04917-19/07/17 | Van-Mai Cao-Lormeau |
| Horizon 2020 - Research and Innovation Framework Programme | ZIKAlliance grant 734548 | Van-Mai Cao-Lormeau |
| Investissement d'Avenir Program | Labex IBEID grant ANR-10-LABX-62- IBEID | Jessica Vanhomwegen Jean-Claude Manuguerra |
| Wellcome | 107778/Z/15/Z | Jimmy Whitworth |
| Wellcome | 206250/Z/17/Z | Adam J Kucharski |
| Medical Research Council | MR/J003999/1 | Conall H Watson |
| National Health and Medical Research Council | 1109035 | Colleen L Lau |
| Medical Research Council | MR/N013638/1 | Alasdair D Henderson |
| Ministry of Europe and Foreign Affairs | Pacific funds 02918-18/06/18 | Van-Mai Cao-Lormeau |

| French and French Polynesia government MA'I'ORE program | 03298/MTF/REC-17/05/18 | Van-Mai Cao-Lormeau |
| French and French Polynesia government MA'I'ORE program | HC/372/DIE/BPT- 18/05/18 | Van-Mai Cao-Lormeau |
| French and French Polynesia government MA'I'ORE program | HC/245/DIE/BPT-16/05/19 | Van-Mai Cao-Lormeau |

The funders had no role in study design, data collection and interpretation, or the decision to submit the work for publication.

## Author contributions
Alasdair D Henderson, Maite Aubry, Data curation, Formal analysis, Writing—original draft; Mike Kama, Anita Teissier, Conall H Watson, Colleen L Lau, Data curation, Formal analysis, Writing—review and editing; Jessica Vanhomwegen, Jean-Claude Manuguerra, Methodology, Writing—review and editing; Teheipuaura Mariteragi-Helle, Tuterarii Paoaafaite, John Edmunds, Jimmy Whitworth, Formal analysis, Writing—review and editing; Yoann Teissier, Data curation, Writing—review and editing; Van-Mai Cao-Lormeau, Adam J Kucharski, Conceptualization, Data curation, Formal analysis, Writing—original draft

## Author ORCIDs
Jean-Claude Manuguerra (iD) http://orcid.org/0000-0002-5202-6531
Adam J Kucharski (iD) https://orcid.org/0000-0001-8814-9421

## Ethics
Human subjects: The French Polynesia serosurveys were approved by the Ethics Committee of French Polynesia (ref 61/CEPF 08/29/2013, 60/CEPF 06/27/2013, 74/CEPF 05/04/2018, and 75/CEPF 05/04/2018). The 2013 Fiji study, and the 2015 and 2017 follow up studies were approved by the Fiji National Research Ethics Review Committee (ref 2013-03, 2015.111.C.D, 2017.20.MC) and the London School of Hygiene and Tropical Medicine Observational Research Ethics Committee (ref 6344, 10207, 12037).

## Decision letter and Author response
Decision letter https://doi.org/10.7554/eLife.48460.sa1
Author response https://doi.org/10.7554/eLife.48460.sa2

# Additional files

## Supplementary files
• Transparent reporting form

## Data availability
All data and code required to reproduce the analysis are available at: https://github.com/a-henderson91/zika-sero-pacific (copy archived at https://github.com/elifesciences-publications/zika-sero-pacific).

The following datasets were generated:

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
