## [Decision Letter]

Thank you for submitting your article "Zika seroprevalence declines and neutralizing antibodies wane in adults following outbreaks in French Polynesia and Fiji" for consideration by *eLife*. Your article has been reviewed by three peer reviewers, including Isabel Rodriguez-Barraquer as the Reviewing Editor and Reviewer #1, and the evaluation has been overseen by Neil Ferguson as the Senior Editor. The following individual involved in review of your submission has agreed to reveal their identity:); Leah Katzelnick (Reviewer #3).

The reviewers have discussed the reviews with one another and the Reviewing Editor has drafted this decision to help you prepare a revised submission.

Summary:

Henderson and Aubry et al. have performed a comparative study of cross-sectional and longitudinal serosurveys conducted before, soon after, and multiple years after the arrival of ZIKV to French Polynesia and Fiji. They use multiple serological assays to measure ZIKV-specific antibodies and DENV-specific antibodies. Specifically, they find an overall decline in ZIKV-specific seroprevalence by ~2 years after the outbreaks in each country, which was primarily driven by a decline in adults (defined as >16 years of age). Children appeared to maintain seroprevalence. They also use longitudinal data from a small subset of individuals from Fiji to show waning in neutralizing antibody titers as well.

These results are worth publishing, but we have several concerns about their presentation and interpretation. In particular, it is not clear to us whether the waning captured by the ELISA/MIA assay is meaningful (as evidence of waning protection) or simply reflects the kinetics of the specific antibody response (anti-domain III) measured by this assay. Data from neutralization assays, the gold standard, is unfortunately scarce (*n* = 45), hard to interpret and not very conclusive because of the timing of sample collection (some waning is expected) and because the evidence of seroreversion is quite weak (only observed in four individuals with weak seroconversions).

Essential revisions:

1) The authors speculate that the reductions in ZIKV seroprevalence measured by ELISA/MIA could indicate waning population immunity to ZIKV. This would have profound implications for future ZIKV dynamics in these and other populations, as the authors discuss. However, we think that another possibility, that is more likely, is that the observed decreases might just reflect the kinetics of antibody responses measured by this specific assay (antibodies against domain III). As far as we understand, it hasn't yet been established whether this antibody response is a good marker of historical exposure, or just a marker of recent exposure. DENV seroprevalence also seems to wane (Supplementary Figure 1). In the absence of additional data describing the performance/kinetics of this assay, this potential explanation needs to be discussed.

2) More details should be provided about the serological assays used. Two references are provided about the ELISA/MIA assays but they also provide very little information. At the very least, the paper should be explicit that the assay uses recombinant antigens comprising domain III of the envelope glycoprotein. Ideally, data on performance characteristics of the assay should also be reported as this information is crucial for the interpretation of results.

3) Related to the point above, please include more detail on the agreement between the MIA and ELISA tests. As far as I can see reference 11 does not contain enough information to assess this. It states: "80% were positive by both tests". Were all the rest of the samples negative by both tests?

4) While ZIKV seroprevalence is observed to wane in both populations (Figure 1), DENV seroprevalence dynamics differ between the populations. It is not clear why, and how this relates to the observed decay for ZIKV seroprevalence. The results seem to be affected by the historical intensity of DENV transmission in French Polynesia and Fiji and may have been affected by DENV or ZIKV transmission that occurred after the main ZIKV epidemics. We recommend that Supplementary Figure 1 accompany Figure 1 in the main text. In Fiji, following the ZIKV epidemic, titers to DENV appeared to increase (except to DENV3). In French Polynesia, there was no DENV immunity to the four DENV before the arrival of ZIKV, and DENV titers did decline in adults after the ZIKV epidemic. Basically, it would be helpful to the reader if the authors include a figure on the epidemics of DENV and ZIKV in these populations before and after the ZIKV epidemic, ideally with serotype-specific epidemic curves if such data are available, to be able to interpret these results.

5) The age-standardised seroprevalence is the important metric here. As the authors note there is less decline in this metric. This should be the main result, as it seems that the other comparisons are not informative at all-as far as I can determine the differences are simply driven by the age distributions of the population sampled. Please explain if this is not the case.

Longitudinal data from Fiji tested using neutralization assays could certainly have added strength to the hypothesis of waning population immunity, but unfortunately the data is scarce (n = 45) and hard to interpret for multiple reasons:

6) In order to be able to compare the DENV and ZIKV results (Figure 2), it would be crucial to understand when sample collection occurred with respect to ZIKV and DENV outbreaks. At the end of the Results, from what we understand: the DENV outbreak was before the ZIKV outbreak, so the observed differences in decline rates between DENV and ZIKV could be explained by much of the DENV decline already happened before the measurement, whereas the ZIKV transmission was much closer to the time of the first measurement. Differences of just a few months could make a large difference here. This would be consistent with the lower increase observed from 2013-2015 for DENV compared to ZIKV. Similarly for the differences in seroreversions, there may have been individuals that were DENV positive closer to the transmission, but were not by 2015, and therefore would not have been captured by the measurements here. In addition, it would be useful to know if there was any DENV transmission in the 2015-2017 period?

7) Also, looking at the neutralization data in GitHub, it seems that 9/45 (20%) of individuals had positive (but low) titers to ZIKV in 2013 (titer greater of equal to 2). 7/9 of these positive values are in individuals who also have positive titers to 3 or more dengue serotypes. We find this concerning as this is supposed to be the pre-Zika time-point and this might suggest some cross-reactivity in this ZIKV PRNT assay. Has the sensitivity and specificity of this neutralization assay (and this cut-point) been calculated? More details need to be provided about this assay and its performance as well.

8) Related to the above, it would be good to show the correlation between neutralization and MIA results. I tried to do this myself (using the provided ids) but was only able to match the data of 16/45 individuals (the neut dataset includes 344 individuals, most with incomplete data). Unless there's a problem with the data (or provided IDs), the agreement seems poor. For example, these 16 individuals have positive neutralization titers in 2015 (>4 in the majority) and yet, only 5/16 are classified as positive by the MIA assay.

9) The four observed ZIKV seroreversions described in the last paragraph of the Results seem to be in individuals with relatively low titers at seroconversion (4). There are four additional "seroreversions" among individuals who were already positive in the 2013 sample. Are these seroreversions mainly capturing cross-reactive responses? It would be useful to show a figure (maybe supplementary) with the longitudinal neutralization titer data.

10) In light of these limitations/uncertainties, we think the manuscript needs to be reworked to emphasize the uncertainty regarding the meaning of these findings. We would emphasize the need for more studies using longitudinal data (rather than cross-sectional) data and a broader set of assays, including PRNTs. We would de-emphasize the discussion/conclusions around the implications of waning protection (last paragraph of the Discussion). While provocative, we are not convinced it's supported by the presented data.

11) It would be good to clarify which subset of samples were tested and the sample sizes that went into each analysis.

[Editors' note: further revisions were requested prior to acceptance, as described below.]

Thank you for submitting your article "Zika seroprevalence declines and neutralizing antibodies wane in adults following outbreaks in French Polynesia and Fiji" for consideration by *eLife*. Your article has been reviewed by a Reviewing Editor and Neil Ferguson as the Senior Editor.

The Reviewing Editor has drafted this decision to help you prepare a revised submission.

The authors have satisfactorily addressed most of our concerns. Thanks for providing the merged data! We have some additional comments:

1) The low agreement reported for the ZIKV MIA and PRNTs assays is a bit concerning, and not consistent with the sensitivity and specificity reported. There seem to be 65 samples positive according to the PRNT in 2017, and of these only 17 (26%) are positive according to the MIA. Assuming that the PRNT is the "gold standard", this would suggest a much lower sensitivity of MIA than was reported for old infections, suggesting time-dependent sensitivity (26% vs. 80%). If instead we question the performance of the PRNT (cross-reactivity?) then the waning results are also questionable. I think these discrepancies (and the difficulty interpreting results from these novel assays) should be explicitly discussed.

2) Related to the above, authors should explicitly state in the text that, while neutralization titers decayed over the observation period, there's little (or no) evidence of sero-reversion (decreasing seroprevalence) according to the neutralization data.

3) Please provide a reference for the sensitivities and specificities reported for the MIA assay. Also, please state whether these performances were evaluated using acute/early convalescent samples vs. samples from historical infections (infections occurring one or more years before).

4) I also find it concerning that 9/46 samples (~20%) were positive against ZIKV (low titers) according to the PRNT assay in 2013. The authors perform some analyses around cross-reactivity, but these are only described in the Materials and methods. I think cross-reactivity (and how it could impact the results) should be addressed in the Discussion.

[Editors' note: further revisions were requested prior to acceptance, as described below.]

Thank you for resubmitting your work entitled "Zika seroprevalence declines and neutralizing antibodies wane in adults following outbreaks in French Polynesia and Fiji" for further consideration by *eLife*. Your revised article has been evaluated by Neil Ferguson (Senior Editor) and a Reviewing Editor.

Most of our concerns have been satisfactorily addressed, but I have one remaining question.

- You report that the proportion seropositive (by MIA) was ~24% in Fiji in 2015, yet Figure 3B shows a seroprevalence of >90% among the 45 individuals that were also tested using neutralization assays. It could almost be seen as if individuals were selected for neutralization assays based on the MIA status (42/45 samples positive by MIA were tested) but the text reads that they were selected based on sample availability only. Please clarify how the samples were selected.

---

## [Author Response]

Summary:Henderson and Aubry et al. have performed a comparative study of cross-sectional and longitudinal serosurveys conducted before, soon after, and multiple years after the arrival of ZIKV to French Polynesia and Fiji. They use multiple serological assays to measure ZIKV-specific antibodies and DENV-specific antibodies. Specifically, they find an overall decline in ZIKV-specific seroprevalence by ~2 years after the outbreaks in each country, which was primarily driven by a decline in adults (defined as >16 years of age). Children appeared to maintain seroprevalence. They also use longitudinal data from a small subset of individuals from Fiji to show waning in neutralizing antibody titers as well.These results are worth publishing, but we have several concerns about their presentation and interpretation. In particular, it is not clear to us whether the waning captured by the ELISA/MIA assay is meaningful (as evidence of waning protection) or simply reflects the kinetics of the specific antibody response (anti-domain III) measured by this assay. Data from neutralization assays, the gold standard, is unfortunately scarce (n=45), hard to interpret and not very conclusive because of the timing of sample collection (some waning is expected) and because the evidence of seroreversion is quite weak (only observed in four individuals with weak seroconversions).

Thank you for the positive view of the overall value of the results. Having read the reviewer comments, we appreciate that the manuscript would benefit from some additional clarifications and analyses, and these are described in our responses below.

Essential revisions:1) The authors speculate that the reductions in ZIKV seroprevalence measured by ELISA/MIA could indicate waning population immunity to ZIKV. This would have profound implications for future ZIKV dynamics in these and other populations, as the authors discuss. However, we think that another possibility, that is more likely, is that the observed decreases might just reflect the kinetics of antibody responses measured by this specific assay (antibodies against domain III). As far as we understand, it hasn't yet been established whether this antibody response is a good marker of historical exposure, or just a marker of recent exposure. DENV seroprevalence also seems to wane (Supplementary Figure 1). In the absence of additional data describing the performance/kinetics of this assay, this potential explanation needs to be discussed.

There is evidence that MIA can provide information about historical exposure, because high seroprevalence was observed in school children in French Polynesia in 2018, four years after the Zika epidemic (Figure 1). We also observed a higher level of seroprevalence against DENV serotypes in older groups in Fiji in 2013, consistent with a long-lasting response. However, we agree it is worth clarifying the antibody responses against domain III are not necessarily equivalent to protective immunity, and now include this in the Discussion:

“Although we found evidence of a decline in seroprevalence for antibodies against domain III of the envelope glycoprotein, as well as waning neutralizing antibody responses following two ZIKV outbreaks, the implications for susceptibility to future ZIKV infection remain unclear. […] A key aim for future work will be establish how waning antibody levels as measured by MIA and neutralization assays correspond to protective immunity, and hence susceptibility to reinfection in populations that have already experienced transmission of ZIKV.”

2) More details should be provided about the serological assays used. Two references are provided about the ELISA/MIA assays but they also provide very little information. At the very least, the paper should be explicit that the assay uses recombinant antigens comprising domain III of the envelope glycoprotein. Ideally, data on performance characteristics of the assay should also be reported as this information is crucial for the interpretation of results.

We have added these details to the Materials and methods:

“Serum samples collected from blood donors between July 2011 and October 2013 and samples collected from the general population and schoolchildren in 2014 were all tested for presence of IgG antibodies against ZIKV and each of the four DENV serotypes using a recombinant antigen-based indirect ELISA as reported previously

[Aubry et al., 2017; Aubry et al., 2015]. […] Recombinant antigens used in both assays comprised domain III of the envelope glycoprotein of ZIKV, DENV-1, DENV-2, DENV-3, or DENV-4 strains (respective GenBank accession no. KJ776791,

AF226686.1, FM986654, FJ44740.1, FM986672.1) and were produced using the

*Drosophila* S2 expression system (Life Technologies, USA) as previously detailed [Aubry et al., 2015].”

3) Related to the point above, please include more detail on the agreement between the MIA and ELISA tests. As far as I can see reference 11 does not contain enough information to assess this. It states: "80% were positive by both tests". Were all the rest of the samples negative by both tests?

We have added additional details to the Materials and methods:

"The sensitivity and specificity of the MIA assay were respectively 100% and 100% for DENV-1, 89.47% and 97.1% for DENV-2, 100% and 100% for DENV-3, 96.88% and 100% for DENV-4, and 79.59% and 94.87% for ZIKV. In the serosurvey conducted among the general population of the 5 archipelagos in French Polynesia in 2014 [Aubry et al., 2017], 196 samples were tested using both ELISA and MIA: among the 97 serum samples that tested positive for anti-ZIKV IgG by ELISA, 78 (80%) were also found positive by MIA; and among the 99 serum samples that tested negative for anti-ZIKV IgG by ELISA, 70 (71%) were also found negative by MIA. This produced a value of Cohen’s κ = 0.51 [Aubry et al., 2017]."

4) While ZIKV seroprevalence is observed to wane in both populations (Figure 1), DENV seroprevalence dynamics differ between the populations. It is not clear why, and how this relates to the observed decay for ZIKV seroprevalence. The results seem to be affected by the historical intensity of DENV transmission in French Polynesia and Fiji and may have been affected by DENV or ZIKV transmission that occurred after the main ZIKV epidemics. We recommend that Supplementary Figure 1 accompany Figure 1 in the main text. In Fiji, following the ZIKV epidemic, titers to DENV appeared to increase (except to DENV3). In French Polynesia, there was no DENV immunity to the four DENV before the arrival of ZIKV, and DENV titers did decline in adults after the ZIKV epidemic. Basically, it would be helpful to the reader if the authors include a figure on the epidemics of DENV and ZIKV in these populations before and after the ZIKV epidemic, ideally with serotype-specific epidemic curves if such data are available, to be able to interpret these results.

We agree it would be useful to include epidemiological context and have updated Figure 1 to include data on which serotypes were confirmed by PCR during this period, as well as temporal trends in symptom reporting. Note that in Supplementary Figure 1 (now Figure 1—figure supplements 1–4), the plots were only intended to show DENV data for 2014 and 2015 in French Polynesia; the apparent zeros in 2013 were the result of redundant plotting code. However, we realise that it would be helpful to show the 2013 data as well, as have updated the figures accordingly.

We also cover the advantages and challenges of the epidemiology in these locations in the updated Discussion:

“These data provided strong evidence that ZIKV seroprevalence declined over the two-year period following first reports of circulation, but our sample size was insufficient to fully explore the potential effect of anti-DENV pre-existing immunity on anti-ZIKV antibody waning once we stratified individuals by previous DENV exposure. Although the outbreaks of DENV-3 in Fiji and ZIKV in French Polynesia were well-documented and occurred over a relatively brief period of time (Figure 1), it was harder to identify the likely time of infection for other viruses – such as ZIKV in Fiji or DENV in French Polynesia – in our study populations.”

5) The age-standardised seroprevalence is the important metric here. As the authors note there is less decline in this metric. This should be the main result, as it seems that the other comparisons are not informative at all-as far as I can determine the differences are simply driven by the age distributions of the population sampled. Please explain if this is not the case.

We think it is important to show the results of age-standardisation, and now do so in Table 2 in the main text. However, we disagree that differences observed are simply the results of the age distributions sampled.

There are three main reasons for this:

First, the age distributions of the 2014 and 2015 study populations were not particularly different (Table 7); both had a similar median and IQR.

Second, age-adjusted DENV seroprevalence estimates for over 16s in French Polynesia were similar to the raw seroprevalence values (now shown in Table 2). Standardising for age increased uncertainty, but did not consistently shift estimates in either direction for these other viruses, suggesting no evidence of a systematic bias as a result of sampling.

Third, ZIKV risk seems to be higher in younger age groups; in French Polynesia, seroprevalence was higher in school children. Hence the 2015 study, which had a slightly younger median age (43 vs. 47), would if anything be expected to have a raw seroprevalence estimate that was biased upwards rather than downwards.

It is therefore likely that any changes in ZIKV seroprevalence estimates after age standardisation are the result of increased uncertainty owing to relatively low levels of seropositivity in adults, particularly in 2015, and small sample size in 2014, rather than a genuine age effect.

It is worth noting that we have identified some differences between the 2014 and 2015 study populations, particularly in terms of exposure risk. We examined the potential impact of these differences by bootstrap resampling the data according to prior exposure history (see response to point 15 below). We still observe a decline in ZIKV seroprevalence, but no decline for recently circulating DENV serotypes.

Longitudinal data from Fiji tested using neutralization assays could certainly have added strength to the hypothesis of waning population immunity, but unfortunately the data is scarce (n=45) and hard to interpret for multiple reasons:6) In order to be able to compare the DENV and ZIKV results (Figure 2), it would be crucial to understand when sample collection occurred with respect to ZIKV and DENV outbreaks. At the end of the Results, from what we understand: the DENV outbreak was before the ZIKV outbreak, so the observed differences in decline rates between DENV and ZIKV could be explained by much of the DENV decline already happened before the measurement, whereas the ZIKV transmission was much closer to the time of the first measurement. Differences of just a few months could make a large difference here. This would be consistent with the lower increase observed from 2013-2015 for DENV compared to ZIKV. Similarly for the differences in seroreversions, there may have been individuals that were DENV positive closer to the transmission, but were not by 2015, and therefore would not have been captured by the measurements here. In addition, it would be useful to know if there was any DENV transmission in the 2015-2017 period?

We agree it is worth including more details about the potential relationship between epidemiological dynamics and sample collection timing. We now include information on flavivirus circulation in the new Figure 1 (see also response to point 4 above). Although the DENV-3 epidemic Fiji occurred around 18 months for the 2015 sample collection, there was also low-level circulation of DENV-1 and DENV-2 during this period. We therefore analysed log neutralisation titres for participants who were seronegative to these respective viruses in 2013 and seroconverted in 2015. Unlike ZIKV, we found no evidence of a subsequent overall decline during 2015–17 (Figure 2—figure supplement 2). We describe this analysis in the updated Results section:

“In Fiji, there was a delay of around 18 months between the end of the 2013/14 DENV-3 epidemic and collection of samples in 2015. […] Unlike ZIKV, we found no evidence of a subsequent overall decline during 2015–17 for those participants who seroconverted during 2013–15 (Figure 2—figure supplement 2).”

7) Also, looking at the neutralization data in GitHub, it seems that 9/45 (20%) of individuals had positive (but low) titers to ZIKV in 2013 (titer greater of equal to 2). 7/9 of these positive values are in individuals who also have positive titers to 3 or more dengue serotypes. We find this concerning as this is supposed to be the pre-Zika time-point and this might suggest some cross-reactivity in this ZIKV PRNT assay. Has the sensitivity and specificity of this neutralization assay (and this cut-point) been calculated? More details need to be provided about this assay and its performance as well.

The sensitivity and specificity of this assay using this cut-off has not been calculated. However, log-titres to ZIKV follow a bimodal distribution (Figure 3 in new manuscript), which supports a cut-off of 2 as an appropriate definition of seropositivity. We also examined the correlation between MIA and neutralisation assay results, finding 68–75% agreement between the two (see response to point 8 below).

As log titres may have been influenced by cross-reactivity, we also compared correlations in responses against different pairs of viruses. For participants who were seronegative to all five viruses in 2013, we found no evidence of cross-reactive responses against ZIKV and the four DENV serotypes (Figure 3—figure supplement 2 in new manuscript), although there was evidence of potential cross-reaction for participants who were already seropositive to DENV (Figure 3—figure supplement 3 in new manuscript). In our main analysis using neutralisation titres (Figure 2), we therefore conditioned on participants being seronegative to ZIKV in 2013.

These additional analyses of neutralisation data are now described in the ‘Serological analysis: Fiji’ section of the Materials and methods.

8) Related to the above, it would be good to show the correlation between neutralization and MIA results. I tried to do this myself (using the provided ids) but was only able to match the data of 16/45 individuals (the neut dataset includes 344 individuals, most with incomplete data). Unless there's a problem with the data (or provided IDs), the agreement seems poor. For example, these 16 individuals have positive neutralization titers in 2015 (>4 in the majority) and yet, only 5/16 are classified as positive by the MIA assay.

The IDs in the two datasets provided in the original submission were independent, as the datasets were not analysed together. In the new submission, we include a single merged dataset (“dset5-fiji-mergedassaydata.csv”), and use this to assess the correlation between neutralization and MIA results. This is described in the updated Materials and methods:

“A previous study, which tested serological samples from Fiji across three divisions [Kama et al., 2019], found that of the samples reactive by MIA, 66/83 (79.5%) exhibited neutralizing activity for ZIKV (κ = 0.71) and 109/112 (97.3%) for DENV (κ = 0.80). […] We found that 54/79 (68.4%) samples that were positive to ZIKV in the neutralization assay were also positive by MIA, and 42/56 (75%) who were seronegative were also negative by MIA.”

While compiling the linked dataset, we also realised that two participants had multiple aliquots tested for MIA. We have therefore omitted these repeat MIA measurements from the dataset, leaving 189 unique samples. We have updated all the values in the text accordingly, and the overall conclusions remain the same.

9) The four observed ZIKV seroreversions described in the last paragraph of the Results seem to be in individuals with relatively low titers at seroconversion (4). There are four additional "seroreversions" among individuals who were already positive in the 2013 sample. Are these seroreversions mainly capturing cross-reactive responses? It would be useful to show a figure (maybe supplementary) with the longitudinal neutralization titer data.

Given the limited neutralisation results we had available, we aimed to focus on the overall titre dynamics, rather than focusing too heavily on results from simple cut-off metrics, which as suggested, may not fully reflect the range of the underlying responses. The main reason we conditioned on seronegativity in Figure 2 was to remove potential effects of cross-reactivity responses in 2013 (see response to point 7 above). We realise, however, that it would also be helpful to show the same analysis including participants who were already seropositive by neutralisation assay to ZIKV or DENV in 2013. We therefore have added a new figure (Figure 2—figure supplement 1) showing results for participants who had a rise in log titre of at least 2 between 2013–15, regardless of titre in 2013. We obtain the same conclusion about waning, actually with a larger average decline in log titre for ZIKV (panel A). This is further evidence that individuals can exhibit a notable decline in ZIKV response, even if they do not fall below the titre = 2 threshold.

Thank you also for the suggestion to show the raw titres. We agree this would be useful and have added this to the new manuscript (Figure 3—figure supplement 1).

10) In light of these limitations/uncertainties, we think the manuscript needs to be reworked to emphasize the uncertainty regarding the meaning of these findings. We would emphasize the need for more studies using longitudinal data (rather than cross-sectional) data and a broader set of assays, including PRNTs. We would de-emphasize the discussion/conclusions around the implications of waning protection (last paragraph of the Discussion). While provocative, we are not convinced it's supported by the presented data.

We have de-emphasised the potential implications for protective immunity in the Abstract and Discussion, and focus instead on the need for additional studies to address the questions raised and remaining. The final paragraph of the Discussion is now as follows:

“In the short-term, our findings have implications for the design of follow up studies of ZIKV. […] Such studies will be essential to understand different aspects of the short and long-term immune antibody response against ZIKV, and how prior exposures to DENV may influence these responses.”

11) It would be good to clarify which subset of samples were tested and the sample sizes that went into each analysis.

We have added the sample sizes at relevant points in the text, as well as to figure and table legends.

[Editors' note: further revisions were requested prior to acceptance, as described below.]The authors have satisfactorily addressed most of our concerns. Thanks for providing the merged data! We have some additional comments:1) The low agreement reported for the ZIKV MIA and PRNTs assays is a bit concerning, and not consistent with the sensitivity and specificity reported. There seem to be 65 samples positive according to the PRNT in 2017, and of these only 17 (26%) are positive according to the MIA. Assuming that the PRNT is the "gold standard", this would suggest a much lower sensitivity of MIA than was reported for old infections, suggesting time-dependent sensitivity (26% vs. 80%). If instead we question the performance of the PRNT (cross-reactivity?) then the waning results are also questionable. I think these discrepancies (and the difficulty interpreting results from these novel assays) should be explicitly discussed.

As MIA and NT measure different aspects of the immune response, we decided to analyse the two results in more detail. First, we compared ZIKV seroprevalence estimates according to pre-existing DENV titre in 2013 (new Figure 3). As already noted in the previous iteration of the manuscript, we found some evidence of cross-reactivity by NT for participants with higher initial DENV titres, which is why we conditioned on participants being seronegative by NT in 2013 in our main analysis of the titre data (Figure 2). We found little difference between assays in 2015. The difference in 2017 was greatest for participants with some pre-existing DENV titre, consistent with the B cell competition hypothesis presented in previous version of the Discussion.

These findings are described in the updated Results:

“Of the 45 participants tested by neutralization assay, 9 were initially seropositive to ZIKV by NT in 2013. […] This difference was associated with participants’ 2013 DENV titres: those with intermediate DENV titres in 2013 had a significantly lower probability of being seropositive in the MIA in 2017 compared to NT (Figure 3C).”

We have also expanded the hypothesis about immunological mechanism for these findings in the Discussion, proposing an explanation for why MIA responses did not persist:

“If an individual has experienced prior DENV infections, high numbers of weakly neutralizing cross-reactive B cells may outcompete naïve B cells for ZIKV antigen [Midgley et al., 2011], leading to a short-term boost in antibody response against ZIKV following ZIKV infection [Robbiani et al., 2017] but not a persistent specific response; a similar phenomenon has been observed for other antigenically variable viruses like influenza [Kucharski et al., 20181]. […] This difference was greatest for participants who had intermediate baseline titres to DENV in 2013 (Figure 3C), which would support the hypothesis that prior DENV exposure may result in a detectable short-term specific response against ZIKV following ZIKV infection (as measured by MIA), but not a persistent specific response.”

2) Related to the above, authors should explicitly state in the text that, while neutralization titers decayed over the observation period, there's little (or no) evidence of sero-reversion (decreasing seroprevalence) according to the neutralization data.

There is some limited evidence of sero-reversion according to neutralization titres, and we now include the numbers in the Results:

“In total, four participants seroreverted between 2015 and 2017; all had a log titre of 4 against ZIKV in 2015.”

3) Please provide a reference for the sensitivities and specificities reported for the MIA assay. Also, please state whether these performances were evaluated using acute/early convalescent samples vs. samples from historical infections (infections occurring one or more years before).

We now clarify in the Materials and methods that serostatus was defined by a cut-off determined using positive and negative control sera analyzed by ROC curve, with the quoted sensitivities and specificities for MIA calculated from these results. The positive control serum for ZIKV was collected 13 months after RT-PCR confirmed infection.

In the revised Discussion, we also note that the post-infection immune response in RT-PCR-confirmed cases may not be the same as in asymptomatic or unreported cases:

“antibody responses in RT-PCR-confirmed cases may not necessarily be representative of immune responses against ZIKV in the wider population, particularly following asymptomatic infection. Although MIA seropositivity in our study was defined using control sera collected over a year after RT-PCR-confirmed infection, our results suggest that this threshold may not detect long-term waning responses in individuals who had unreported, and likely less severe, infections.”

4) I also find it concerning that 9/46 samples (~20%) were positive against ZIKV (low titers) according to the PRNT assay in 2013. The authors perform some analyses around cross-reactivity, but these are only described in the Materials and methods. I think cross-reactivity (and how it could impact the results) should be addressed in the Discussion.

As noted in the response to point 1 above, participants who were seropositive by NT in 2013 typically had higher initial DENV titres, suggesting potential cross-reactivity. However, our main analysis focused on participants who were seronegative to ZIKV in 2013, so this should not affect our conclusions.

We include the additional analysis as Figure 3 in the new Results and address the issue of cross-reactivity in the updated Discussion:

“Several participants in Fiji were seropositive to ZIKV by neutralization assay (NT) in 2013, but this result may be influenced by cross-reaction; participants who had high pre-existing titres to DENV in 2013 were more likely to be seropositive by NT (Figure 3A). […] However, we obtained the same conclusion when participants who were initially seropositive were also considered (Figure 2—figure supplement 1).”

[Editors' note: further revisions were requested prior to acceptance, as described below.]Most of our concerns have been satisfactorily addressed, but I have one remaining question.- You report that the proportion seropositive (by MIA) was ~24% in Fiji in 2015, yet Figure 3B shows a seroprevalence of >90% among the 45 individuals that were also tested using neutralization assays. It could almost be seen as if individuals were selected for neutralization assays based on the MIA status (42/45 samples positive by MIA were tested) but the text reads that they were selected based on sample availability only. Please clarify how the samples were selected.

Thank you for raising this point, and we realise we should have been clearer here. The NT testing was indeed informed by the 2013 and 2015 MIA status, as well as sample availability (i.e. from loss to follow up in the longitudinal serosurvey and stored serum availability). However, this selection choice did not affect our conclusions about waning of ZIKV titres, because our analysis focused on individual-level changes in NT titre conditioned on serostatus, rather than absolute seroprevalence by NT.

We have updated the Materials and methods accordingly:

“To follow the evolution of antibody titres at the individual level, a subset of samples collected from the same individuals in 2013, 2015 and 2017 were tested for the presence of neutralizing antibodies against ZIKV and each of the four DENV serotypes using a neutralization assay as previously described [Cao-Lormeau et al., 2016]. […] We also tested samples from an additional 24 participants from the same cohort who were seropositive to ZIKV by MIA in 2013 or 2015 and for whom we had sufficient serum from 2013 and 2015 to test by neutralization assay, but no matched sample from the 2017 follow up survey (i.e. 69 paired samples in total).”

As well as the Results:

“To explore dynamics of antibody waning at the individual level, we performed neutralization assays (NT) on a subset of 45 participants from Fiji for whom sufficient sera were available to test against ZIKV from all three collection periods, focusing on those who were seropositive to ZIKV by MIA in 2013 or 2015.”